# Self-Medication for the Treatment of Abdominal Cramps and Pain—A Real-Life Comparison of Three Frequently Used Preparations

**DOI:** 10.3390/jcm11216361

**Published:** 2022-10-28

**Authors:** Martin Storr, Harald Weigmann, Sabine Landes, Martin C. Michel

**Affiliations:** 1Center of Endoscopy, 82319 Starnberg, Germany; 2Medical Clinic II, Ludwig-Maximilians-University, 81377 Munich, Germany; 3Consumer Health Care, Sanofi Germany, 65926 Frankfurt, Germany; 4Department of Pharmacology, University Medical Center, Johannes Gutenberg University, 55131 Mainz, Germany

**Keywords:** gastrointestinal cramps, gastrointestinal pain, irritable bowel syndrome, hyoscine butyl bromide, peppermint oil, over-the-counter treatment, pharmacy-based patient survey

## Abstract

Functional gastrointestinal disorders (FGIDs), including irritable bowel syndrome (IBS), are frequently handled by self-management with over-the-counter (OTC) products such as hyoscine butylbromide (HBB), alone or in combination with paracetamol, and natural products such as peppermint oil. To obtain real-world information, we have performed an anonymous pharmacy-based patient survey among 1686 users of HBB, HBB + paracetamol, and peppermint oil. Based on the distinct but overlapping indications for the three OTC products, multiple logistic regression was applied to compare them in users reporting gastrointestinal cramps and pain, bloating, flatulence, or IBS as cardinal symptoms. All three treatments reduced symptoms and associated impairments of work/daily chores, leisure activities, and sleep by approximately 50%. Based on the four cardinal symptoms and the four dependent continuous variables of interest (change in intensity of symptoms and of the three impairment domains) a total of 16 logistic regression models were applied. HBB, HBB + paracetamol, and peppermint oil had similar reported overall effectiveness in those models. Gender, age, baseline symptom severity, and impairment in one of three domains had small and inconsistent effects on perceived treatment success. We provide evidence that HBB, HBB + paracetamol, and peppermint oil have comparable effectiveness in their approved indications under real-world conditions in an OTC setting.

## 1. Introduction

Functional gastrointestinal disorders (FGIDs) are a frequent cause of gastrointestinal (GI) cramps and pain, bloating, and flatulence. FGIDs have been reported to be present in up to 30% of the population in Western societies [1,2]. Symptoms of FGIDs can occur singularly or in various combinations [3] and in the absence or presence of changes in stool texture such as diarrhea and/or constipation may meet the Rome IV criteria of irritable bowel syndrome (IBS) [4].

In the absence of red flags, FGIDs are frequently handled by self-management with prescription-free, over-the-counter (OTC) products [5]. Various OTC products are available for the treatment of FGID symptoms, and patient preferences may differ between countries [6]. OTC options include the muscarinic receptor antagonist hyoscine butylbromide (HBB; also known as butylscopolamine bromide or scopolamine butylbromide) and natural products such as peppermint oil (PO). While OTC products differ in molecular targets, their functional targets are similar or at least largely overlapping. For instance, both HBB and PO have spasmolytic effects and are recommended for the treatment of IBS symptoms, but the former acts as a muscarinic receptor antagonist [7], and the latter by inhibiting Ca^2+^ channels [8]. The effectiveness of HBB in the treatment of GI cramps and pain can be further improved by the addition of paracetamol (also known as acetaminophen) [9], and fixed-dose combinations of HBB plus paracetamol are available (PLUS).

Approved OTC indications for the mentioned drugs and drug combinations are distinct but overlapping. For instance, HBB has been placed on the list of indispensable drugs by the World Health Organization. While approved indications differ by formulation and jurisdiction, it is approved for instance in Germany for the treatment of mild to moderate spasms of the GI tract as well as of spasmodic abdominal pain in IBS. PLUS is approved for the relief of spasmodic pain in GI conditions and of spasmodic pain in functional disorders in the biliary tract, the lower urinary tract, and the female genital system such as in dysmenorrhea in adults and adolescents aged 12 years and older. On the other hand, PO is approved for the symptomatic relief of abdominal pain, minor spasms of the GI tract, and flatulence, especially in patients with IBS in adults and adolescents aged 12 years and older and weighing at least 40 kg.

Direct comparisons of the effectiveness and tolerability of such medications for the symptomatic management of abdominal cramps and/or pain are lacking. Given that these preparations are available as OTC products, randomized controlled trials comparing them may have limited external validity in a real-world setting [10]. Therefore, we have conducted a pharmacy-based patient survey (PBPS) to compare three products with the above-mentioned active ingredients, a dragée containing HBB, a film-coated tablet containing PLUS, and a soft capsule containing PO. Given that patients use these three products for overlapping but distinct indications/cardinal symptoms, we have compared them within an intended use using applied logistic regression analyses; these regression models also included possible confounding factors involved in the effectiveness of these products within a shared indication/cardinal symptom.

## 2. Materials and Methods

### 2.1. Tested Products

Three products with overlapping indications in the therapeutic area of FGIDs involving GI cramps and pain were investigated, which are available in German pharmacies as OTC products. These were a dragée containing 10 mg of HBB (Buscopan^®^ Dragée, A. Nattermann & Cie GmbH, Frankfurt, Germany), a film-coated PLUS tablet containing a fixed-dose combination of 10 mg HBB plus 500 mg paracetamol (Buscopan^®^ plus, A. Nattermann & Cie GmbH), and a soft capsule containing 0.2 mL PO (corresponding to 181.6 mg *Mentha piperita* L., aetheroleum; Buscomint^®^ bei Reizdarm, A. Nattermann & Cie GmbH).

### 2.2. Study Design

A non-interventional, prospective PBPS was conducted via community and online pharmacies in Germany between 1 July 2020 and 31 June 2021. Patients who had purchased one of the three branded medicines and consented to participate were handed out a questionnaire. The questionnaire was to be filled out at the participants’ discretion and to be sent in a pre-stamped envelope to a contract research organization (Winicker Norimed GmbH, Nuremberg, Germany) for the collection and analysis of the data. Inclusion criteria were the purchase of one of the three products, willingness and ability to fill out the paper-based study questionnaire, and age of ≥18 years. There were no further exclusion criteria. The survey was anonymous, and no information was collected allowing post hoc identification of participants. Applicable German laws and regulations neither required nor recommended the involvement of an ethical committee for an anonymous survey such as the one reported here; this is in line with other recently reported PBPS from Germany [10,11,12,13,14].

Recruitment of approximately 1000 participants per product from 200 pharmacies (community and online) was intended. Due to slower than targeted recruitment during the COVID-19 pandemic, the recruitment was stopped prematurely after a total of 1704 questionnaires were received; 18 questionnaires were excluded from analysis due to lack of information on age and/or use of the product, resulting in 1686 eligible subjects recruited via 110 pharmacies. The three preparations were represented by roughly similar numbers of participants (HBB 579, PLUS 641, PO 466). Community and online pharmacies contributed similar numbers of questionnaires (881 and 805, respectively).

The questionnaires for the three products were comparable in content and contained questions on demographic variables (gender, age), history of GI symptoms in the past 30 days, the perceived trigger of current complaints (selection from a list of possible triggers), and the timespan of conditions. However, this was different between them: The HBB and PLUS questionnaires focused on the timespan of pain/complaints before treatment initiation, i.e., proposing the categories after 30 min, 60 min, 2 h, or more than 2 h, whereas the PO questionnaires focused on the timespan of the condition, i.e., proposing the options of <1, 1–3, 3–6, 6–24 months or >2 years. The current condition was captured by asking questions on the intensity of pain/complaints prior to first ingestions, the cardinal symptom triggering current use (GI cramps and pain, IBS, bloating, flatulence) for HBB and PO, and GI cramps and pain, lower urinary tract complaints such as urinary tract infection, dysmenorrhea or other for PLUS, the impact of current condition on work/daily chores, leisure activities and sleep. The questions on the severity of pain/complaints and on associated impact were rated on an 11-point Likert scale (0–10 from no pain/complaint/impact to very strong pain/complaint/impact); all other questions were asked categorically. The questions on the intensity of pain/complaint and on the impact of current conditions, and were repeated after drug intake (1 h after the first intake for HBB and PLUS, 4 h after intake for PO). The questionnaires asked about the onset of relief following first ingestion were asked: the available options were 0–5, 6–15, 16–30, 31–45, 46–60, and >60 min for HBB and PLUS and <1, 1–2, 2–3, 3–4 and >4 h for PO. Finally, perceived general effectiveness, tolerability, and treatment satisfaction were captured.

### 2.3. Data Analysis

All analyses were performed based on a statistical analysis plan developed by the authors. The plan for primary analyses had been finalized before any data were viewed. After primary analysis, a post hoc statistical analysis plan was developed for secondary analyses using logistic regression analyses. Secondary analysis compared continuous variables between groups of patients using different preparations for one cardinal symptom, i.e., GI cramps and pain for all three products, and IBS, bloating, and flatulence for the HBB and PO groups. To avoid artifacts based on small sample sizes, secondary analyses were limited to preparation/cardinal symptom combinations with available data on at least 100 subjects. The models used treatment, gender, age, and baseline values of the symptom, impairment of work/daily chores, leisure activities, and sleep as independent variables. Dependent variables were a change in symptom and a change in impairment. Baseline values were not included as independent variables if a change in the same parameter was the dependent variable. A patient was included in each model for which he/she had reported the corresponding cardinal symptom. Effect sizes for parameters with a descriptive *p*-value < 0.05 are reported with the respective 95% confidence interval (CI).

All calculations were performed by Winicker Norimed GmbH using SAS (version 9.2, Cary, NC, USA). In line with recent guidelines for enhanced robustness of data analysis [15,16], we considered all data to be exploratory. Therefore, as recommended by leading statisticians [17], we do not report *p*-values and focus on effect sizes with a presentation of 95% CI. In the absence of statistical hypothesis testing, we partly report on possible differences between groups based on qualitative evaluation such as “comparable” or the weaker “largely comparable”.

## 3. Results

### 3.1. Patient Characteristics

Demographic information is provided in Table 1. Users of the three preparations were of comparable age (mean 44–46 years) and comparable gender distribution (78–84% female); the slightly higher share of females in the PLUS group may relate to the fact that this is the only product also approved for use in dysmenorrhea. The number of days with GI cramps and pain, bloating, and/or flatulence in the past month was comparable for HBB and PLUS, and considerably greater with PO, possibly reflecting the different intended use. Among present symptoms that led to the purchase of the product, GI cramps and pain were highest in the HBB group (85.8% compared with approximately 60% in the other groups), whereas IBS, bloating, and flatulence were more frequent in the PO compared with the other groups; urinary tract complaints and dysmenorrhea were only reported in the plus group (Table 1). These differences reflect the approved indications of the respective product.

### 3.2. Pre-Treatment Symptoms

Perceived triggers of the current complaints were largely comparable among users of the three preparations (Appendix A). Stress, nutrition, and bloating were indicated most frequently. The intensity of pain/complaints was comparable in all groups of participants, i.e., ranging around 7 on a 0–10 Likert scale (Figure 1). Similarly, participants in all three groups reported a comparable degree of impairment in work/daily chores, leisure activities, and sleep (Figure 2). Across all three preparations, impairment of work/daily chores and leisure activities was rated stronger compared with that of sleep (mean rating of ≈6 vs. ≈4). Interestingly, users of HBB and PLUS were largely repeated users (75.3% and 79.2%, respectively), whereas those of PO were largely first-time users (69.4%). The latter may be explained because PO had been on the market only 1 year at the time the survey was performed.

### 3.3. Treatment Outcomes

The start of treatment relative to the onset of symptoms was comparable for HBB and PLUS (75.0% and 78.6% within the first hour). This question was not asked from PO users due to the more chronic nature of IBS and IBS symptoms; however, users of PO reported that the first drug administration was several months after the start of symptoms (17.5% after <1 month, 26.3% after 1–3 months, 15.9% after 3–6 months, 17.0% after 6–24 months and 23.3% >24 months).

Users of all three preparations experienced a reduction in the intensity of GI cramps and pain rated more than 50%, i.e., from ≈7 to ≈3 on a 0–10 Likert scale (Figure 1). The percentage of patients reporting a reduction of at least 50% was comparable across preparations (68.8% for HBB, 73.4% for PLUS, 62.2% for PO). The degree of impairment of work/daily chores, leisure activities, and sleep was reduced by more than 50% with all three preparations (Figure 2). The onset of symptom relief occurred within 60 min following administration of the first dose in 93.9% of HBB users and in 93.2% of PLUS users (in more than 50% within 6–30 min). In contrast, symptom relief occurred in 20.1% of PO users within 1 h and in 91.2% within 4 h.

For the global effectiveness rating, 97.2% of HBB users, 96.4% of PLUS users, and 86.1% of PO users were satisfied or very satisfied with the effectiveness of the treatment. Global tolerability was ranked as good or very good by 99.0%, 97.1%, and 92.0% of HBB, PLUS, and PO users, respectively. In the global rating of satisfaction with treatment, 97.2%, 96.1%, and 85.8% of HBB, PLUS, and PO users, respectively, were very satisfied or satisfied. No adverse events were reported.

### 3.4. Post Hoc Logistic Regression Analyses

Based on the four cardinal symptoms (GI cramps and pain for all three products; IBS, bloating, and flatulence for HBB and PO) and the four dependent continuous variables of interest (change in intensity of symptom and of impairments) a total of 16 regression models were performed to evaluate factors associated with improvement. Findings for changes in symptom intensity are shown in Table 2, and those for all other outcome variables are shown in Appendix A. For the dependent variable improvement of symptom intensity of the cardinal symptom GI cramps and pain, preparation, age, impairment of leisure activities and of sleep at baseline had CI, not including 0 in the logistic regression analysis: Compared with the reference group PLUS, PO was slightly less effective (−0.5949 [CI −0.9697; −0.2202] units) whereas HBB exhibited similar effectiveness as PLUS. For each higher year of life, improvement was 0.0106 [−0.0186; −0.0025] units lower, whereas participants with greater impairment of leisure activities or of sleep at baseline had a greater improvement of symptom severity (leisure activities: 0.1757 [0.0788; 0.2726] units per unit of baseline score on a 0–10 scale; sleep: 0.0731 [0.0236; 0.1277]. Among the total of all 16 models (Table 2, Appendix A), preparation had a CI not including 0 twice, gender once, age four times, baseline symptom severity five times, baseline impairment of work/daily chores five times, of leisure activities 11 times, and of sleep four times. Of note, the effect size estimate was small in all cases (less than 10%, mostly less than 5%; c.f. Table 2, Appendix A) compared with the overall reduction in symptom intensity or impairment (Figure 1 and Figure 2), indicating that the explanatory independent variables including preparation used had only limited effects on treatment-associated improvements.

## 4. Discussion

### 4.1. Critique of Methods

The present study reports analyses based on a PBPS, which implies specific strengths and weaknesses that need to be discussed prior to the interpretation of the findings. Randomized controlled trials are considered the gold standard of evidence generation in clinical medicine. HBB [7,9,18,19,20,21,22,23,24], PLUS [9] and PO [8,25] have proven effectiveness in controlled trials, compared with a placebo for each of them. However, based on the trial setting and on the inclusion and exclusion criteria such randomized, controlled trials are not helpful to reflect real-world conditions in OTC settings where they are inherently dispensed without the involvement of a prescribing physician [26]. On the other hand, PBPS are established tools in the evaluation of OTC products and were used in real-world trials in various indications including functional bowel disorders [10], headache and migraine [11,14,27], or cough and cold [12,13]. While PBPS are not suitable to prove the effectiveness or tolerability of a drug in absolute terms, such proof was not intended in our study since proof of effectiveness had been obtained in the above-cited controlled trials. Thus, PBPS are a useful tool to gather real-world information on OTC drugs. The absence of a control group implies that PBPS should not be used to derive claims on absolute effectiveness or tolerability; however, PBPS directly comparing multiple OTC products may be used to obtain information on their relative effectiveness and tolerability under real-world conditions.

The questionnaire did not ask for comorbidities, comedications, or for concomitant use of more than one of the products under investigation; while the former is likely to exist in some patients, we consider the latter to be unlikely. Lack of knowledge on possible comedications, particularly the differential prevalence of such groups is a possible limitation of the study.

HBB, PLUS, and PO are all approved OTC products for the treatment of GI cramps and pain. Since each of them has multiple approved indications, we asked patients to name the cardinal symptom for which they had purchased the product. Offered cardinal symptom options corresponded to approved indications and reflect the way OTC products are intended to be used.

The presence of multiple distinct but overlapping approved uses for HBB, PLUS and PO has important implications for the interpretation of our data. Some cardinal symptoms and their corresponding approved uses are limited to some products. For instance, all three are approved for GI cramps and pain; however, PLUS is not approved for IBS, bloating, and flatulence, but is the only preparation approved for lower urinary tract symptoms and dysmenorrhea. On the other hand, a user may have more than one symptom motivating their purchase of the product. This reflects the observation that GI symptoms can occur individually or in variable combinations [3]. These factors leave it challenging to define cohorts with matched symptoms to enable comparison across treatment options. However, comprising subgroups, these limitations can be addressed by logistic regression analyses as performed in the present study.

The present study was designed prior to the COVID-19 pandemic and conducted while the pandemic was ongoing, largely prior to the broad availability of vaccines. This substantially impacted patient care for medical conditions and led to poor recruitment in clinical trials across various indications, frequently leading to pausing or even discontinuation of trials [28]. The present study had recruited 56% of the intended maximal number of participants after one year, when investigators decided to stop recruitment since 1686 eligible participants at that time were considered a sufficient number. Of note, the originally intended number of participants was not based on formal sample size calculations due to the exploratory character of the study. The exploratory character of our study had implications for our approach to data analysis. In line with recent recommendations [15,16,17], we did not calculate *p*-values but reported effect sizes with their respective CI.

### 4.2. Comparison of Populations

The users of the three preparations were comparable in general demographics except for a slightly greater share of females for PLUS, which may be explained by PLUS being the only product approved for dysmenorrhea. PO users, on average, had more days with GI cramps and pain, bloating or flatulence in the past month, which is in line with the observed presence of these symptoms (and of IBS) being twice as high and reflecting that IBS is one main indication for PO. Bloating and flatulence were also mentioned more frequently by PO users compared with the other two products as triggers for the current GI cramps and pain. Interestingly, the most frequently mentioned trigger for all three medicines was stress followed by food intake (e.g., too fat or too sweet). Of note, these are patient perceptions, which do not necessarily represent established cause-effect relationships. These distinct differences in cardinal symptoms/indications confirm the overlapping complaints of the three user groups but also show that they vary to some extent. This observation triggered the decision for post hoc logistic regression analyses for exploratory comparisons between the preparations within patient groups defined by one of the symptoms (see below).

### 4.3. Perceived Effectiveness and Tolerability

All three treatments roughly halved the intensity of pain/complaints and of impairments due to symptoms on a 0–10 point Likert scale (Figure 1 and Figure 2). Moreover, 62–73% of users reported a reduction of at least 50%. This is well in agreement with the proven effectiveness of HBB [7], PLUS [9], and PO [8,25], all established in comparison to placebo in randomized controlled trials. However, these and other studies have also reported a markable placebo effect in the treatment of FGIDs. The share of the observed responses attributable to the placebo or the verum effect cannot be determined from our present data, a feature intrinsic to real-world-evidence studies in general and to PBPS in particular. However, the present data provides an impression on the overall degree of symptom relief that patients can expect from use of the OTC products tested here. Thus, the overall effect gives us an impression on the effectiveness of the compounds in real life, rather than the effectiveness compared with a placebo. More importantly, they provide initial insight into their relative effectiveness in the respective indications (for details see below).

Previous studies in FGIDs have reported that patients consider the time to the onset of symptom relief to be important for their satisfaction with treatment [6]. In the present study, more than 90% of HBB or PLUS users reported the onset of symptom relief within 1 h and more than 50% within 30 min. Symptom relief was slower in PO users with 20% reporting it within 1 h and 90% within 4 h. Such differences may be explained by different symptoms being treated and by pharmacokinetic differences between the preparations. HBB is a quaternary amine with poor systemic bioavailability [7] and is likely to directly act on muscarinic receptors in the gut [29] causing a rapid onset of action. In contrast, menthol (an active ingredient of PO) in its stomach-resistant capsule formulation reaches peak plasma concentrations approximately 5 h after oral administration [30]. Interestingly, the alignment of perceived time-to-onset of symptom relief and of pharmacokinetic aspects provides circumstantial additional evidence that the reported effects largely represent pharmacodynamic action and to a minor extent a placebo effect.

Our results report a global effectiveness rating of ≥95% as very good or good for HBB and PLUS and ≥85% with PO. On the other hand, all three preparations had a global tolerability rating of good to very good in ≥90% of participants. The combination of high effectiveness and a good to very good tolerability led to reported global satisfaction with treatment rating of satisfying to very satisfying in ≥95% of HBB and PLUS users and ≥85% in PO users, the latter possibly reflecting that treatment of IBS remains a medical challenge and that IBS patients often look back to a considerable journey of treatment attempts that has left them with a more critical view of treatments.

### 4.4. Comparison of Preparations and Exploration of Other Explanatory Variables

To compare preparations within an indication and to adjust for potential confounding factors such as gender, age, baseline symptom, and impairment intensity, we have performed logistic regression analyses. Based on possible combinations of cardinal symptom and outcome parameters, a total of 16 models were analyzed (Table 2, Appendix A). The factor most consistently associated with outcome was baseline impairment of leisure activities (11 times in 12 models), whereas basal symptoms and basal impairment of work/daily chores and of sleep were statistically identifiable factors in only 6, 5, and 4 out of 12 models, respectively. Interestingly, the four models identifying sleep as an associated factor, all had improvement of pain/symptom rating as the dependent variable. Of note, baseline values were not included in the models that had improvement of the same parameter as the dependent variable, which excludes regression to the mean as a relevant factor. Gender was identified only once and age only four times as explanatory variables in 16 models. Most importantly, preparation was identified only twice across 16 models as an identifiable independent variable with these being the indications of GI cramps and pain and of IBS for the outcome parameter of pain/symptom improvement. While overall improvement in pain/symptom was about 3.5 points, parameter estimates for a smaller improvement by PO relative to the reference group were 0.5949–0.7166, i.e., contributing in a minor way only to the therapeutic success of the treatments.

## 5. Conclusions

We conclude that HBB, PLUS, and PO, OTC treatments targeting abdominal symptoms, have high patient-reported effectiveness in relieving cardinal symptoms such as abdominal cramps and pain, IBS, bloating, and flatulence. These findings extend knowledge from placebo or active compound controlled randomized trials for the three preparations to a real-world-evidence setting. Logistic regression models indicate that gender and age have little effect on effectiveness, making all three compounds valuable OTC options in all age groups, a finding not yet documented in randomized clinical trials. The most consistent factors associated with later treatment success are the baseline impairment of leisure activities, work/daily chores, sleep, and basal symptom severity. Within the approved use, the personal choice of preparations has limited and inconsistent effects on effectiveness. Associated with a reported good to very good tolerability, this finally results in a high degree of patient satisfaction with these treatments.

## Figures and Tables

**Figure 1 jcm-11-06361-f001:**
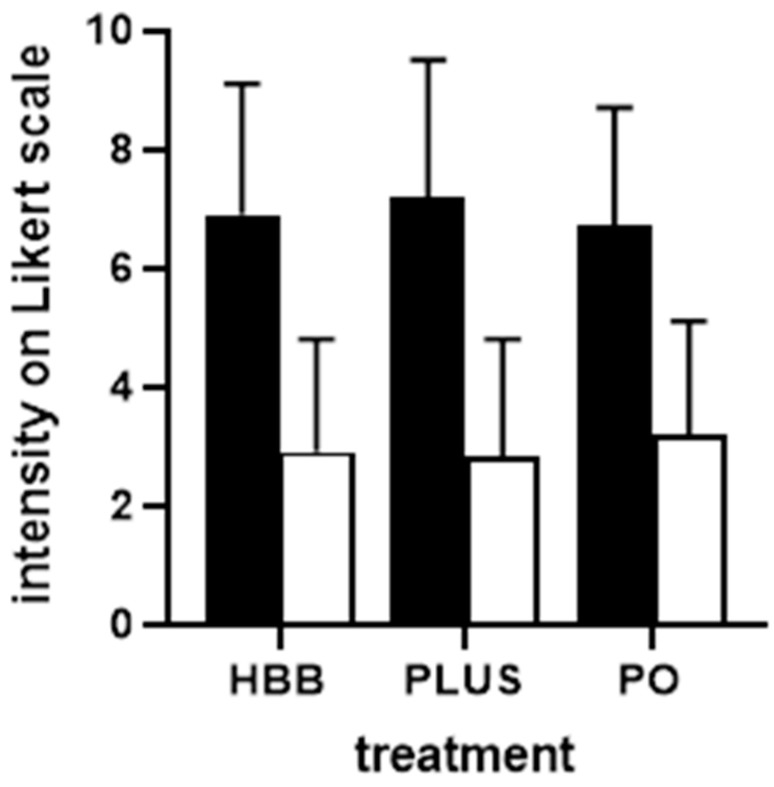
Intensity of GI cramps and pain before (filled bars) and after 1st dose of treatment (open bars) in users of the three preparations. Data are shown as means ± SD. Note that the after-treatment data were collected 1 h after intake for HBB and PLUS and after 4 h for PO. HBB, hyoscine butylbromide, PLUS, HBB + paracetamol; PO; peppermint oil.

**Figure 2 jcm-11-06361-f002:**
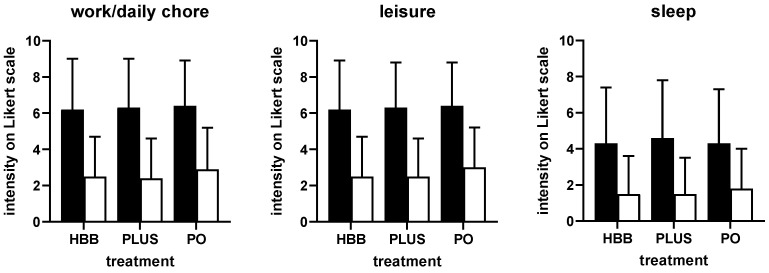
Impairment of work/daily chores, leisure activities, and sleep before (filled bars) and after 1st dose of treatment (open bars) in users of the three preparations. Data are shown as means ± SD. HBB, hyoscine butylbromide, PLUS, HBB + paracetamol; PO; peppermint oil.

**Table 1 jcm-11-06361-t001:** Demographics and past and current complaints of survey participants. Age and complaints in past 30 days are shown as means ± SD. Gender and current complaints are shown as % of users. Multiple nominations possible for complaints were possible. n.o.: not offered as checkbox in line with approved indication. HBB, hyoscine butylbromide, PLUS, HBB + paracetamol; PO; peppermint oil; GI, gastrointestinal; IBS, irritative bowel syndrome.

	HBB	PLUS	PO
Demographics			
n	579	641	466
Age, years	45.9 ± 18.6	43.8 ± 17.8	46.0 ± 16.2
Gender, % female	77.8	84.2	78.8
Complaints in past 30 days			
GI cramps and pain, number of days	6.6 ± 6.2	5.9 ± 5.6	10.3 ± 7.6
Bloating, number of days	7.5 ± 7.7	6.1 ± 6.8	11.3 ± 8.4
Flatulence, number of days	4.8 ± 8.4	6.9 ± 7.8	11.7 ± 8.7
Current complaints			
GI cramps and pain, %	85.8	59.9	60.5
Urinary tract complaints	n.o.	9.6	n.o.
Dysmenorrhea	n.o.	51.4	n.o.
IBS	22.5	n.o.	62.9
Bloating	37.4	n.o.	62.9
Flatulence	20.4	n.o.	49.6
Other	16.4	6.6	3.9

**Table 2 jcm-11-06361-t002:** Effects of preparation, demographic, and baseline variables on improvement in symptom score on a 0–10 Likert scale, which was by 4.0, 3.4, and 3.5 points for HBB, PLUS, and PO, respectively, in the overall group (see Figure 1). Data are shown as effect size estimates with 95% CI in 1128, 381, 502, and 343 patients, respectively. These are absolute effect sizes for the categorical independent variables (preparations and gender) relative to the indicated reference group; they are values per year of age and per score of baseline values on the Likert scale for the continuous independent variables. n.d.: not determined because not an approved indication for the preparation. HBB, hyoscine butylbromide, PLUS, HBB + paracetamol; PO; peppermint oil; GI, gastrointestinal; IBS, irritative bowel syndrome.

	Indication			
	GI Cramps and Pain	IBS	Bloating	Flatulence
Preparation				
HBB	−0.1562 [−0.4824; 0.1701]	reference group	reference group	reference group
PLUS	reference group	n.d.	n.d.	n.d.
PO	−0.5949 [−0.9697; −0.2202]	−0.7166 [−1.2096; −0.2237]	−0.2013 [−0.6088; 0.2062]	0.0094 [−0.4829; 0.5016]
Gender				
Female	reference group	reference group	reference group	reference group
Male	−0.1785 [−0.5105; 0.1536]	−0.1465 [−0.6867; 0.37379	0.0015 [−0.4706; 0.4736]	−0.0488 [−0.5907; 0.4930]
Age	−0.0106 [−0.0186; −0.0025]	−0.0208 [−0.0348; −0.0068]	−0.0129 [−0.0255; 0.0004]	−0.0075 [−0.0215; 0.0065]
Baseline work/chore impairment	0.0319 [−0.0608; 0.1245]	0.0400 [−0.1176; 0.1976]	0.0233 [−0.1222; 0.1688]	0.0333 [−0.1285; 0.1952]
Baseline leisure impairment	0.1757 [0.0788; 0.2706]	0.2589 [0.0875; 0.1976]	0.2272 [0.0732; 0.3811]	0.2264 [0.0516; 0.4011]
Baseline sleep impairment	0.073 [0.0236; 0.1227]	0.1009 [0.0217; 0.1800]	0.1123 [0.0374; 0.1872]	0.0929 [0.0049; 0.1808]

## Data Availability

Data are owned by Sanofi Consumer Health Care (Frankfurt, Germany). They will be made available to qualified non-commercial investigators upon request to the corresponding author.

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
