# Peer review of "Self-Medication for the Treatment of Abdominal Cramps and Pain—A Real-Life Comparison of Three Frequently Used Preparations"

_jcm, 2022, doi:10.3390/jcm11216361_

Round 1

Reviewer 1 Report

Thank you for the opportunity to review this manuscript. This study provided concrete understanding the impact of three OTC products on abdominal cramps and pain. The study aims to sound relatively straight forward though I do think that some revision is needed. 

Abstract, this study investigated three products (HBB Buscopan® Dragée, HBB PLUS Buscopan® plus, PO Buscomint® bei Reizdarm), not three medications (HBB, HBB + paracetamol and peppermint oil), may need to clarify since there may be other forms or products of these three medications.

Please also clarify if patients are allowed to take any of these three products (overlapping) or allowed to take only one product each time.  

Purchasing the medications did not warrant taking the medications. Please revise the inclusion criteria accordingly.

Page 4 line 163-164, supplemental table 1, what does “largely comparable” mean? If comparison analysis was run, please report the results. If not, please run the analysis and report the results.

Page 4 line 166, please check “0-10 Likert scale (Figure 1)”, most likely, a 0-10 scale for pain assessment is not a Likert scale, i.e., 0 means no pain/complaints and 10 means the worst pain/complaints imaginable.

Page 4 line 169, line 170, and Page 5 line 198, stronger comparable, mean rating of ≈6 vs. ≈4 Please be aware of the words of scientific writing. May report median or report the means and standard deviation.

May add note of p value of the comparisons in figure 1 and 2.

Page 6 line 216-217, please be aware of the sentence “Findings for symptom intensity are shown in Table 2, all others are shown in Supplemental Tables 2-4.”. It seems that the authors compared the effects of three products on reducing symptom intensities, not the effects of those products on symptom intensity.

May add the sample size for each of the 16 regression models. How the patients with two or more cardinal symptoms were treated? i.e., a patient reported GI cramps and pain, and bloating.

Page 8 line 318-323, The way the symptom improvement recorded may also impact the results, i.e., was the symptom improvement reported by patients’ recall or by a timer?

Page 9 line 366-369, these two sentences seem unclear.

Author Response

Reviewer comment: Thank you for the opportunity to review this manuscript. This study provided concrete understanding the impact of three OTC products on abdominal cramps and pain. The study aims to sound relatively straight forward though I do think that some revision is needed.

Author reply: We appreciate the overall positive evaluation by the referee.

Reviewer comment: Abstract, this study investigated three products (HBB Buscopan® Dragée, HBB PLUS Buscopan® plus, PO Buscomint® bei Reizdarm), not three medications (HBB, HBB + paracetamol and peppermint oil), may need to clarify since there may be other forms or products of these three medications.

Author reply: We fully agree with the referee that we have studied products, not medications. However, the Abstract of the original submission had already consistently talked about products, not medications. The only point where the main manuscript talks about medications is in the main Introduction, where we do not refer to the three specific products under investigation but to medications in general that have the same ingredients. Therefore, no modification was made.

Reviewer comment: Please also clarify if patients are allowed to take any of these three products (overlapping) or allowed to take only one product each time. Purchasing the medications did not warrant taking the medications. Please revise the inclusion criteria accordingly.

Author reply: We do not have specific information whether patients used more than one of three products under investigation or whether and which comedications were used. This is a possible limitation of the present study. However, this does not change the inclusion criteria we had described. To address this, we have added the following wording as 2nd paragraph of section 4.1 (l. 261-265).

“The questionnaire did not ask for comorbidities, comedications or for concomitant use of more than one of the products under investigation; while the former is likely to exist in some patients, we consider the latter to be unlikely. Lack of knowledge on possible comedications, particularly differential prevalence of such groups is a possible limitation of the study.”

Reviewer comment: Page 4 line 163-164, supplemental table 1, what does “largely comparable” mean? If comparison analysis was run, please report the results. If not, please run the analysis and report the results.

Author reply: In line with recommendations from leading physicians (see e.g., ref. 15-17), the fact that one of the authors (MCM) is the responsible professor for statistics courses in his and other institutions, and the exploratory character of our study, we use p-values only sparingly and focus on effect sizes and whether their magnitude was deemed to be clinically relevant. Thus, our study protocol had pre-specified which measures were to be compared statistically. For all others, we consider it inappropriate to conduct and report such analyses post-hoc. In this spirit we use more qualitative wording such as “comparable” or (weaker) “largely comparably”. This is explained in the revised manuscript at the end of section 2.3 (l. 139-141).

Reviewer comment: Page 4 line 166, please check “0-10 Likert scale (Figure 1)”, most likely, a 0-10 scale for pain assessment is not a Likert scale, i.e., 0 means no pain/complaints and 10 means the worst pain/complaints imaginable.

Author reply: The scale offered to the participants is a continuous scale for which the two ends are defined as “no complaints” and as “worst pain/complaints possible”, but no intermediate steps were defined. In our view this constitutes a Likert scale (see e.g., https://en.wikipedia.org/wiki/Likert_scale).

Reviewer comment: Page 4 line 169, line 170, and Page 5 line 198, stronger comparable, mean rating of ≈6 vs. ≈4 Please be aware of the words of scientific writing. May report median or report the means and standard deviation.

Author reply: We assume that the referee refers to the text relative to Figures 1-2. However, there may be a misunderstanding. Of course, we do not consider numerical differences of about ≈6 vs. ≈4 to be comparable. However, these are the differences between before and after drug intake, not the differences between products. Thus, the groups at baseline and the groups after drug intake are comparable, whereas values before and after drug intake are not.

Reviewer comment: May add note of p value of the comparisons in figure 1 and 2.

Author reply: In line with the recommendations by various guidelines and leading statisticians (references 15-17) and based on the exploratory character of the study, we had explicitly chosen not to perform hypothesis-testing statistics on such before vs. after data. We feel that these data provide a helpful view on medication effects under real-life conditions. However, in the absence of a control group, they should not be abused to derive claims of efficacy. This had been addressed in Methods (l. 135-137) and Discussion (l. 254-257).

Reviewer comment: Page 6 line 216-217, please be aware of the sentence “Findings for symptom intensity are shown in Table 2, all others are shown in Supplemental Tables 2-4.”.

Author reply: We have reworded that sentence for clarity (l. 215-216).

Reviewer comment: It seems that the authors compared the effects of three products on reducing symptom intensities, not the effects of those products on symptom intensity.

Author reply: This is an astute observation. We have reworded this (l. 215).

Reviewer comment: May add the sample size for each of the 16 regression models. How the patients with two or more cardinal symptoms were treated? i.e., a patient reported GI cramps and pain, and bloating.

Author reply: The sample size has been added for each of the 16 models in the corresponding table legends (l. 236 of main paper, l. 3 of each legend of supplement tables 2-4). We also have clarified that a patient was included in each model for which he/she exhibited the cardinal symptom under investigation (l. 132-133).

Reviewer comment: Page 8 line 318-323, The way the symptom improvement recorded may also impact the results, i.e., was the symptom improvement reported by patients’ recall or by a timer?

Author reply: The questionnaire for the participants asked them to fill in baseline symptom/impairment intensity and that 1 h (HBB and PLUS) and 4 h (PO) after intake of medication (see l. 114). Based on the non-interventional character of study, we did not provide instructions beyond that. Therefore, we cannot say whether they fill the questionnaire immediately at those time points or somewhat later based on recall.

Reviewer comment: Page 9 line 366-369, these two sentences seem unclear.

Author reply: We do not understand this comment. L. 366-369 in the submitted manuscript (now l. 376-382) contain the mandatory sentences on supplementary materials and author contributions. If your concern remains, please identify the text that you are concerned about.

Reviewer 2 Report

Dear Authors,

 I have read the manuscript entitled "Self-Medication for the Treatment of Abdominal Cramps and Pain – a Real-Life Comparison of Three Frequently Used Preparations".

The work addresses a very current topic, namely self-medication. Thus, the effectiveness of three OTC products (butyl-scopolamine bromide, its combinations with paracetamol and peppermint oil), used to treat abdominal cramps and gastrointestinal pain, symptoms present in patients with functional gastrointestinal disorders, including irritable bowel syndrome, was evaluated. (IBS).

The study, carried out by means of a questionnaire in various pharmacies in Germany, is very valuable because it is a clinical one that finally included 1,686 users of these OTC products. The very good tolerability and satisfaction of patients who used these OTC products in the treatment of various symptoms, especially pain, are the strong attributes of this study.

The paper is presented in a clear and concise manner, respecting the requirements of the journal. The bibliographic references are in reasonable number, some of them being recent studies. The obtained results were presented in the 2 tables and 2 figures, to which the information from the supplementary materials is added, which allows easy reading of the statistical data. Although they were not mandatory, I appreciate that you also gave us some pertinent conclusions.

However, I have a few minor comments:

1.      The vast knowledge of pharmacokinetics and pharmacodynamics of the authors cannot be doubted. You used the Likert scale as a tool to assess the severity of pain intensity. Can the authors argue why they chose this scale as an assessment tool?

2.      You mentioned in the manuscript that the average age was between 44-46 years. Did the users of these OTCs suffer from other illnesses besides those for which they purchased the products? Were there also patients with polypathologies? If so, do the authors have information about possible polymedication among these patients?

3.      In the event of polymedication when using these OTCs, there is a risk of interactions. Do the authors have information on the use of drugs with analgesic or co-analgesic action in parallel with OTCs? How do we quantify the analgesic effect achieved?

4.      Self-medication is currently a big problem worldwide. The modern patient is tempted to be independent and solve his own health problems. Even though tolerability and satisfaction of treatment were highlighted in this study, however, how do the authors think encouraging OTC use should be approached?

Author Response

Reviewer comment: Dear Authors, I have read the manuscript entitled "Self-Medication for the Treatment of Abdominal Cramps and Pain – a Real-Life Comparison of Three Frequently Used Preparations". The work addresses a very current topic, namely self-medication. Thus, the effectiveness of three OTC products (butyl-scopolamine bromide, its combinations with paracetamol and peppermint oil), used to treat abdominal cramps and gastrointestinal pain, symptoms present in patients with functional gastrointestinal disorders, including irritable bowel syndrome, was evaluated. (IBS). The study, carried out by means of a questionnaire in various pharmacies in Germany, is very valuable because it is a clinical one that finally included 1,686 users of these OTC products. The very good tolerability and satisfaction of patients who used these OTC products in the treatment of various symptoms, especially pain, are the strong attributes of this study. The paper is presented in a clear and concise manner, respecting the requirements of the journal. The bibliographic references are in reasonable number, some of them being recent studies. The obtained results were presented in the 2 tables and 2 figures, to which the information from the supplementary materials is added, which allows easy reading of the statistical data. Although they were not mandatory, I appreciate that you also gave us some pertinent conclusions.

Author reply: We appreciate the positive evaluation of our work by the referee.

Reviewer comment: However, I have a few minor comments:

  1. The vast knowledge of pharmacokinetics and pharmacodynamics of the authors cannot be doubted. You used the Likert scale as a tool to assess the severity of pain intensity. Can the authors argue why they chose this scale as an assessment tool?

Author reply: Our scale extended from 0 to 10 and the outer ends were defined as “no complaints” and as “worst pain/complaints possible”, but no intermediate steps were defined. This enables considering the responses to be a continuous variable, which in turn justifies calculating means ± SD. Moreover, this type of scale is easily understood by patients and allows them to quantitatively describe their symptom intensity.

  1. You mentioned in the manuscript that the average age was between 44-46 years. Did the users of these OTCs suffer from other illnesses besides those for which they purchased the products? Were there also patients with polypathologies? If so, do the authors have information about possible polymedication among these patients?

Author reply: The referee raises an interesting point, i.e., whether some of our participants suffered from concomitant conditions and used concomitant medications. Based on the non-interventional character of the study, we had chosen to limit the number of questions being asked. Therefore, we do not have information in this regard. However, we will consider obtaining such information in future studies. To address this, we have added the following wording as 2nd paragraph of section 4.1 (l. 261-265).

“The questionnaire did not ask for comorbidities, comedications or for concomitant use of more than one of the products under investigation; while the former is likely to exist in some patients, we consider the latter to be unlikely. Lack of knowledge on possible comedications, particularly differential prevalence of such groups is a possible limitation of the study.”

  1. In the event of polymedication when using these OTCs, there is a risk of interactions. Do the authors have information on the use of drugs with analgesic or co-analgesic action in parallel with OTCs? How do we quantify the analgesic effect achieved?

Author reply: This comment is a logical extension of the previous one. As explained above, we did not collect such information but consider it interesting to consider it in future studies.

  1. Self-medication is currently a big problem worldwide. The modern patient is tempted to be independent and solve his own health problems. Even though tolerability and satisfaction of treatment were highlighted in this study, however, how do the authors think encouraging OTC use should be approached?

Author reply: The referee raises an interesting health policy question. The use of self-medication has advantages and limitations. Therefore, approval of medications for self-management of conditions is tightly regulated in Germany and many other countries by the appropriate regulatory authorities. A possible disadvantage of self-management by OTC is that patients may make inappropriate decision on whether to take a certain OTC drug, how often, how long and in which doses. Therefore, the regulatory authorities in Germany and many other countries consider whether a given indication and a given medicine is suitable for self-management. The European Union has issued a clear directive in this regard (Directive 2001/83/EC as amended and Human Medicines Regulations 2012). The resulting decision on availability of a certain medicine for self-management is accompanied by a mandate to provide certain information on appropriate and inappropriate uses and related warnings on the package and in the patient information leaflet inside the package. On the other hand, self-medication has clear benefits. It takes some workload off physicians to focus their time on those patients needing diagnostic measures and closer monitoring. Moreover, self-medication empowers patients and transforms them from passive recipients to active participants in the management of their medical conditions. Of course, both of these positive aspects only are applicable for certain conditions and medicine. This being subject to careful regulation by the competent authorities, we feel that self-management can be an important part of the management of certain conditions by certain medications. As none of these considerations is germane to the present study and as our study did not generate any data in this regard, we prefer not to add such discussion to the manuscript.